# Uterine Perforation as a Complication of the Intrauterine Procedures Causing Omentum Incarceration: A Review

**DOI:** 10.3390/diagnostics13020331

**Published:** 2023-01-16

**Authors:** George Lucian Zorilă, Răzvan Grigoraș Căpitănescu, Roxana Cristina Drăgușin, Anca-Maria Istrate-Ofițeru, Elena Bernad, Mădălina Dobie, Sandor Bernad, Marius Craina, Iuliana Ceaușu, Marius Cristian Marinaş, Maria-Cristina Comănescu, Marian Valentin Zorilă, Ileana Drocaș, Elena Iuliana Anamaria Berbecaru, Dominic Gabriel Iliescu

**Affiliations:** 1Department of Obstetrics and Gynaecology, University of Medicine and Pharmacy of Craiova, 200642 Craiova, Romania; 2Department of Obstetrics and Gynaecology, University Emergency County Hospital of Craiova, 200642 Craiova, Romania; 3Department of Obstetrics and Gynaecology, Medgin, GINECHO Clinic, 200333 Craiova, Romania; 4Department of Histology, University of Medicine and Pharmacy of Craiova, 200642 Craiova, Romania; 5Research Centre for Microscopic Morphology and Immunology, University of Medicine and Pharmacy of Craiova, 200642 Craiova, Romania; 6Department of Obstetrics and Gynaecology, “Victor Babes” University of Medicine and Pharmacy, Eftimie Murgu Square no 2, 300041 Timisoara, Romania; 7Clinic of Obstetrics and Gynaecology, “PiusBrinzeu” County Emergency Hospital, 300723 Timisoara, Romania; 8Lugoj Municipal Council, Medical Assistance-Education Service Romania, 305500 Lugoj, Romania; 9Romanian Academy Timisoara Branch, Mihai Viteazul Avenue, 24, 300275 Timisoara, Romania; 10Department of Obstetrics and Gynaecology, “Carol Davila” University of Medicine and Pharmacy, “Dr I. Cantacuzno” Hospital, 020021 Bucharest, Romania; 11Department of Anatomy, University of Medicine and Pharmacy of Craiova, 200642 Craiova, Romania; 12Department of Forensic Medicine, University of Medicine and Pharmacy of Craiova, 200642 Craiova, Romania; 13Doctoral School, University of Medicine and Pharmacy of Craiova, 200642 Craiova, Romania

**Keywords:** intrauterine procedure, dilation & curettage, uterine perforation, pregnancy, abortion, omentum, systematic review

## Abstract

Objective: Omentum involvement resulting from uterine perforation is a rare complication following intrauterine procedures that might require immediate intervention due to severe ischemic consequences. This review examines the prevalence of this complication, risk factors, the mode and timing of diagnosis, the proper management and the outcome. Methods: A systematic literature search was conducted on PubMed, PubMed Central and Scopus using uterine perforation, D&C, abortion and omentum as keywords. The exclusion criteria included the presence of the uterus or placenta’s malignancy and uterine perforation following delivery or caused by an intrauterine device. Results: The review included 11 articles from 133 screened papers. We identified 12 cases that three evaluators further analysed. We also present the case of a 32-year-old woman diagnosed with uterine perforation and omentum involvement. The patient underwent a hysteroscopic procedure with resectioning the protruding omentum into the uterine cavity, followed by intrauterine device insertion. Conclusion: This paper highlights the importance of a comprehensive gynaecological evaluation following a D&C procedure that includes a thorough clinical examination and a detailed ultrasound assessment. Healthcare providers should not overlook the diagnosis of omentum involvement in the presence of a history of intrauterine procedures.

## 1. Introduction

Uterine perforation of both the gravid and the non-gravid uterus is associated with substantial morbidity and sometimes mortality. Cervical dilatation and curettage (D&C) are among the most commonly performed gynaecological procedures worldwide despite being highly invasive. This procedure is most widely used for surgical termination of pregnancy and in various gynaecological conditions for hemostatic, evacuation and biopsy purposes. It is well-known that any intrauterine procedure, from a simple aspiration to a more difficult curettage, involves a risk of uterine perforation [1,2,3]. However, the incidence of uterine perforation has been estimated to be very low, at approximately 0.8–6.4/1000 procedures [4]. It mainly depends on the technique, the healthcare provider’s experience and the risk factors associated with the preexisting medical problem [5,6,7]. Parity, advanced age and general anaesthesia increase the risk of uterine perforation, while uterine retroversion does not significantly contribute [8,9,10,11]. Therefore, in a healthy uterus, perforation can often be misdiagnosed or overlooked, because of the low expectation of this complication, and this may also contribute to the low incidence reported by the current literature [4]. Still, the non-obstetric diagnostic and therapeutic indications for D&C cover a wide spectrum of conditions accompanied by abnormal uterine bleeding, such as endometrial hyperplasia, prolonged heavy menstrual bleeding or postmenopausal bleeding [12,13]. D&C complications include haemorrhage most frequently, while uterine perforation is estimated at 0.3% and 2.6% in premenopausal and postmenopausal women, respectively [14].

Since 2009, the rate of unintended pregnancy and, consequently, surgical termination of pregnancy by D&C has fallen significantly in high-income countries. However, the rate of abortion remains high in low- and middle-income countries [15]. More, unsafe termination of pregnancy causes 8–11% of global maternal deaths. The safety of abortion depends on the equipment used, the health facility and the skilled human resources. Clandestine abortion represents the termination on request of a pregnancy by people without proper medical training and/or in an environment with poor medical standards [16]. Illegal termination of pregnancies is a threat to the health and survival of a female patient and an independent factor in maternal morbidity and mortality [17].

Uterine perforation following D&C can affect pelvic structures/organs and their potential involvement or traction into the uterine cavity [3,18,19,20,21,22,23]. The injury of the surrounding organs can sometimes lead to emergencies that require prompt medical intervention, potentially endangering the patient’s life. One of the rarest but still possible complications is the incarceration of the omentum in the uterine cavity following uterine perforation during an intrauterine procedure. The symptomatology that accompanies this condition is not specific and sometimes inapparent. The timing of a proper diagnosis can sometimes vary between a few hours to a few years from the moment of the manoeuver. To our knowledge, no review regarding uterine perforation after a surgical procedure with omentum incarceration has yet been reported. The purpose of the current research was to examine the incidence, risk factors, clinical presentation, imaging examination and timing from D&C to the correct diagnosis of uterine perforation with omentum incarceration and to evaluate the impact on women’s healthcare.

## 2. Materials and Methods

### 2.1. Study Selection

We conducted a systematic literature search of Pubmed, Pubmed Central and Scopus published between 1 January 1972–30 September 2022, including all available English language full-text articles. We used the following terms: ‘uterine perforation’, ‘dilation and curettage’, ‘abortion’, and ‘omentum’. We restricted all the searches only to human studies. We aimed to investigate the incidence and impact of this condition in general low-risk settings; therefore, we excluded the cases with (1) the presence of the malignancy of the uterus or placenta, (2) uterine perforation after dilatation and curettage after delivery and (3) uterine perforation caused by intrauterine dispositive (IUD). There were three additional records identified through other sources (Figure 1). We decided not to include the conditions that represent risk factors for uterine perforation, because in such cases the professionals are well-aware of the potential complications. Instead, we aimed to describe the diagnosis and outcome of uterine perforation with omentum incarceration in low-risk women, where the expectations for such complications is low and the diagnosis can be easily overlooked.

The studies were examined by two separate researchers (ZGL and ID), who screened the articles and excluded the duplicates in the first stage. Next, abstracts of all potentially relevant papers were individually assessed for suitability. The publications that did not fit the inclusion criteria were rejected. Discussion with a third researcher (EB)helped to reconcile disagreements between the two initial reviewers.

### 2.2. Data Synthesis

The study aimed to investigate the incidence, risk factors, clinical and imaging presentation, and timing from the D&C to the correct diagnosis of uterine perforation with omentum incarceration. We also evaluated the impact on women’s health.

## 3. Results

We identified 134 potentially relevant full-text communications. After the exclusion of one duplicate, 133 screened records were further analysed. Only 21 articles were considered eligible, of which ten were excluded for specific reasons. There were ninecase reports [24,25,26,27,28,29,30,31,32], one case series [33] and one letter to the editor [34] included in the analysis (Table 1).

## 4. Case Report

A 32-year-old patient was referred to our Obstetric-Gynecology Clinic for a potential uterine perforation following pregnancy termination on request 4 h ago. From her medical history, we noted a previous delivery by Cesarean Section three years before for breech presentation and fetal macrosomia. The patient did not report any other pregnancies, miscarriages or abortions on request. Her medical state was excellent, we noted a body mass index of 19.5 and that she was a non-smoker. Her vital signs were normal, with a blood pressure of 110/60 mmHg, a heart rate of 85 beats per minute, 36.6 °C body temperature, and there was no abdominal distension or tenderness during the abdomen examination. 

Conservative management was planned as the patient presented only slight vaginal bleeding and minimal free fluid in the pelvis. We decided on hospitalisation for close surveillance under antibiotic and uterotonics therapy. However, in the longitudinal view of the uterus, we identified an echogenic band in the uterine wall and cavity extending from the uterine fundus to the cervical external os, suggesting possible momentum incarceration. We examined the uterine body’s transversal plane for the echogenic area’s width evaluation, and the 3D reconstruction of the uterine coronal plane showed us the endometrial and cervical cavities with an accurate mapping of the echogenic area (Figure 2). Twenty-four hours after the curettage, the patient was stable and with no clinical symptoms. We decided to discharge her with a reschedule for a hysteroscopic procedure after two weeks. The patient returned for this minimally invasive procedure and we confirmed the omentum incarceration as a fibro-lipomatous appearance string running from the uterine fundus, next to the tubal ostium and continuing through the entire endometrial cavity to the cervical canal (Figure 3). We performed a hysteroscopic resection, and the tissue removed was later confirmed by the pathology exam as omentum. We finished the intervention by inserting an intrauterine device, as the patient had expressed a desire for contraception before the procedure. Antibiotics and anti-inflammatory medication were pursued for the following seven days. The patient returned one month later for a check-up. The scan reconfirmed the correct placement of the intrauterine device with a typical characteristic of the uterine structure.

## 5. Discussion

Uterine perforation represents a potential complication of the intrauterine manoeuvers used for endometrial cavity evacuation or sampling. Although rare, it may determine immediate or distant severe consequences for the patient’s health. In addition, this iatrogenic condition, defined as a perforation and local destruction of the entire uterine wall, can compromise future fertility [14]. Uterine perforation has been reported to be more frequent secondary to an obstetric D&C. It has also been described in cases where a non-obstetric D&C or vacuum aspiration was applied [21,35]. Uterine rupture usually indicates an injury of the uterine wall secondary to a iatrogenic insult [36].

Perforation is considered severe and life-threatening if it leads to immediate heavy bleeding. Therefore, uterine perforation should be suspected in the presence of incontrollable significant bleeding during or after D&C. The symptoms and the severity of uterine perforation are influenced by its uterine location or the presence of an underlying condition, such as a scar pregnancy or uterine cancer. 

### 5.1. Incidence and Risk Factors

Uterine perforation has been documented in roughly 0.3% of premenopausal females and 2.6% of postmenopausal females undergoing D&C for non-pregnancy-related illnesses. The risk of perforation is slightly elevated for the pregnancy-related procedures. It is particularly prevalent (up to 5%) in the cases where the procedure is used to control postpartum hemorrhage. Approximately 0.5% of first- and second-trimester procedures (induced or spontaneous abortions) result in uterine perforation [37]. The actual incidence of uterine perforation with omentum incarceration is unknown and most probably higher than published. This is because of the rare occurrence of instrumental uterine perforation, while an unknown number of cases are not reported and published in the medical literature for liability reasons. Other reasons involve the cases that require immediate intervention in complicated uterine perforations, unrecognized perforations without further complications and investigations and pre-hospital mortality in very low-income countries [2].

There have been reported some conditions and risk factors which can contribute to the occurrence of uterine perforation: problematic dilation of the cervix (primiparous or menopause), scarred cervix after surgical manoeuvers or previous vaginal deliveries, abnormal positions of the uterus (malposition of the uterus), deformations of the uterine cavity due to pathological uterine formations (leiomyoma, adhesions), scarred uterus (previous injury to the uterine wall, last cesarean section), conditions that diminish myometrial strength such as pregnancy, especially multiparity, uterine infections, advanced age, connective tissue disorders such as Ehrler-Danlos and Loeys-Diets syndrome, and the use of general anesthesia [8].

The pelvic structures that can engage in the uterine cavity are the omentum, the appendix, the small bowel, the ovary or the fallopian tube [1,2,18,19,20,22,23].

The present study analysedthe publications wherethe incarceration of the omentum was described due to uterine perforation secondary to an intrauterine manoeuvre. Eleven studies were identified [24,25,26,27,28,29,30,31,32,33,34] that included 12 cases. In all cases, curettage manoeuvres were identified as the cause of the presence of the omentum tissue in the uterine cavity. In 11 patients (91%), D&C or other intrauterine manoeuvres were performed to evacuate a pregnancy in the firstor second trimester by abortion [24,25,26,27,28,29,31,32,33,34] and in one case (8%) the procedure was performed to investigate menopausal bleeding [30]. Unsafe termination of pregnancy was the cause of uterine perforation with subsequent omentum incarceration in two cases.

We could not establish significant risk factors for this complication regarding the traditional circumstances that favor uterine perforation, but we should keep in mind the low number of cases. Most of these patients were ≤30 years old (82%), and only two of them were over 30 years old (18%). Only one case (8%) was an elderly patient in whom the curettage was performed at menopause for diagnostic and therapeutic purposes. In our research, the first gestation and parity did not represent a risk factor, as none of the patients was at their first pregnancy, and only one patient (8%) had no previous deliveries. Regarding the number of deliveries as a potential risk factor, we observed that six patients (50%) had one delivery, four patients (33%) had two deliveries, and only one patient (8%) had three deliveries. Only two cases (16%)had previous delivery by Cesarean section before abortion [26,27]. Twin pregnancy was described in only one case (8%) [24]. Unsafe abortion was noted in two cases (16%) [31,33].

### 5.2. Clinical Presentation

Experienced health providers usually suspect uterine perforation at the time of the dilation and curettage from the loss of resistance during the instrument progression. Moreover, the diagnosis of uterine perforation can be clinically suspected if the patient presents acute abdominal pain, heavy vaginal bleeding or any sign of internal bleeding such as hypotension or tachycardia imagistic detection of peritoneal free fluid. The clinical manifestations can range broadly from mild to severe, depending on the size and cause of the uterine wall injury and related to the location of the perforation most frequent on the body of the uterus, followed by the anterior wall (40%), the cervix (36%) and lastly the fundus of the uterus (13%) [38]. Intraoperative direct visualisation of the breach can confirm the diagnosis. If overlooked, most patients have a good prognosis with spontaneous healing of the uterine perforation. Very few may develop incarceration of the omentum.

There are no reports of specific symptoms that can warn of a potential diagnosis of uterine perforation with omentum incarceration. Our research noted seven cases (58%) that presented with lower abdominal pain (6/7 cases, 85%), while in one case, the patient described severe upper abdominal pain associated with nausea and vomiting (1/7cases, 15%). Four patients (33%) complained of abnormal vaginal bleeding, while one patient(8%) was completely asymptomatic, and one patient mentioned amenorrhea (8%).

Regarding the clinical examination, five of the reviewed case reports (41%) did not mention any data. In four cases (33%), omentum tissue was described coming out of the vagina/introitus or cervical os, while in one patient (8%), the appearanceof a foreign body hanging from the introitus was reported. 

### 5.3. Imaging Examination

A complete diagnosis of uterine perforation with secondary incarceration of the omentum should combine a detailed medical history with a comprehensive clinical examination and an imaging evaluation mainly using ultrasound assessment, but not excluding a computer-tomography (CT), magnetic resonance imaging (MRI) or radiographic evaluation (Table 2). Imaging is essential in patients with a clinical history suggestive of uterine perforation to confirm the myometrial injury and also to investigate the uterine cavity content. The imaging approach can vary based on the institutional guidelines and availability of different equipment and techniques, especially for low-income countries. 

In the Emergency Room, ultrasound is the preferred diagnostic tool to properly assess the regular appearance of the uterus, and uterine perforation can be suspected if there is confirmation of myometrial echogenic appearance of the injury, free fluid in the pelvis or abnormal structures in the endometrial cavity. Thus, the most common imaging features of uterine perforation include heterogenous intrauterine content, hemoperitoneum, pneumoperitoneum and pelvic abscesses [39]. Moreover, ultrasound assessment used routinely to guide intrauterine instruments significantly reduces the risk of uterine perforation. 

The initial imaging modality of choice was ultrasound because it is readily available, cost-effective, free of ionising radiation, and compact mobile machines can be used at the patient’s bedside or inside the operative theatre. A transvaginal approach better assesses the reproductive organs by detecting the perforation site [40,41]. In contrast, a transabdominal approach provides a wider view of the patient’s status, including estimating the volume of the potentially associated hemoperitoneum [42]. A transvaginal ultrasound examination can show the presence of a discontinuity in the uterine serosa with a hyperechoic mass protruding in the wall of the uterine body and cavity extending from the uterine fundus to the cervical external os. This image suggests the presence of the omentum in the uterine cavity. Ultrasonography was the most frequently investigated in six (50%) of the studied cases [24,27,30,31,32,34]. Three-dimensional ultrasound can help the healthcare provider depict the site of the uterine perforation as a hypoechoic or anechoic image in the myometrium or as a track extending from the endometrium to the serosa of the uterus [43]. Because usually there is a decreased perfusion in the uterine wall at the level of the perforation due to the development of a hematoma, in some cases colour Doppler imaging can add information [43].

If ultrasound proves negative or inconclusive, CT can be an adjunct imaging modality that allows the visualisation of all abdominal pelvic organs and diagnosing of pneumoperitoneum [44]. The site of the uterine perforation can be easily assessed using multiplanar reconstructions, while contrast-enhanced CT aids in detecting associated abscesses. When there is a suspicion of associated ureteral and bladder injuries, CT angiography and urography can also identify any affected vessels [45]. However, the role of CT examination in diagnosing uterine perforation with omentum involvement was minimal, as it showed no evidence of bowel injury except hematoma around the perforation scar [25].

The role of MRI is limited to the diagnosis of uterine wall injuries on an urgent basis and is usually used in clinically stable patients and should not delay emergency intervention. However, MRI can aid in challenging cases where ultrasound and CT are not informative, and there is still a high suspicion of uterine perforation [43]. MRI has a superior soft-tissue resolution and can improve the visualisation and identification of uterine perforation with associated complications, such as secondary abscess formation. MRI described a fatty mass in one case (8%), which was useful for diagnosis, along with the ultrasound examination [26].

In patients desiring fertility preservation, catheter angiography can be diagnostic and therapeutic [46]. Uterine arteries embolization can improve overall patient outcomes, as there is no need for a hysterectomy in cases with heavy bleeding secondary to uterine perforation. In addition, catheter angiography with temporary vascular occlusion can be performed even in hemodynamically unstable patients [47,48]. However, many institutions in medium and low-income countries do not provide a 24 h available angiography service. In the current review, we noted no reports of using catheter angiography as a diagnostic and therapeutic tool in patients with uterine perforation and omentum involvement.In certain conditions, such as a previous myomectomy, embolization can cause uterine rupture of the previous scar [49].

Pelvic-abdominal X-rays may be useful in the diagnosis of the uterine perforation [50]. In the study group, an X-ray was used just in one case (8%) to support the diagnosis [28].

### 5.4. Timing of Diagnosis

In four cases (33.3%%), the diagnosis of uterine perforation was confirmed immediately after curettage or established in the next few hours. After birth, two patients (16%) were diagnosed with this rare complication 28 days after the uterine manoeuver, while one (8%) presented unspecific symptoms 17 days later. One patient (8%) was diagnosed with uterine perforation 17 months later, one patient (8%) reported symptoms two years later and one patient (8%) five years later. Thus, we cannot draw a clear conclusion regarding the time omentum incarceration occurs after uterine perforation or when the symptoms develop. 

### 5.5. Management 

When recognised, uterine perforation can be treated conservatively if the patient’s general condition is good, there is no profuse bleeding, and there are no estimated risks related to lesions of the abdominal viscera. Conservative management usually includes hospitalisation, placement of a urinary catheter, antibiotic therapy and vital signs monitoring to detect possible bleeding, peritonitis or intestinal obstruction [51]. An additional evaluation using minimally invasive techniques such as hysteroscopy or laparoscopy can help establish the diagnosis.

Hysteroscopy is a simple tool that allows the gynaecologist to diagnose different uterus disorders, including uterine perforation [52].

Laparoscopy is safe when performed immediately after uterine perforation. A correct diagnosis of the extent of the perforation injury should be obtained before the surgical intervention. Advantages of laparoscopy include a short hospital stay and minimal medico-legal issues [53,54]. Laparotomy is indicated in hemodynamically unstable cases and when extensive instrumentation after perforation has been made or when tissue resembling bowel or fat is confirmed in the uterine cavity [55,56]. 

In our review, three cases (25%) with omentum incarceration after uterine perforation were managed using a hysteroscopic approach. Laparoscopy was performed in five patients (41%), while in two cases (16%), laparoscopy was combined with hysteroscopy and in one case (8%) with hysteroscopy and control cystoscopy. The surgical management involved laparotomy in seven patients (58%) (Table 3).

The most common place of perforation is the uterinefundus, which is also the place where the perforation might be large enough for theomentum and other abdominal organs (intestine, salpinx) to get access to engage into the uterine cavity. One of the myometrial characteristics is contractility, mostly when the uterus has content, which may explain the “absorption” of omentum or intestines even if, initially, the ultrasound shows only the perforation site and anemptycavity after the procedure. The procedure follow-up after a correctly diagnosed uterine perforation, even with a stable hemodynamic patient, should be done systematically in the first 24 h, and alsoafter 1–4 weeks. The presence of any symptom should always triggera complete medical examination to rule out any long-term complications, such as omentum involvement after a uterine perforation [29].

When family planning is complete, permanent sterilisation should be discussed with the patient, as this could prevent the repeating of complications of further intrauterine procedures [14]. In addition, the follow-up should include an ultrasonogram of the uterus and βHCG determination to exclude the possibility of retained products of conception if the uterine perforation followed an obstetric D&C [14].

### 5.6. Outcome 

Patients with uterine perforation usually have good outcomes unless the complication is diagnosed late or there is intraabdominal organ involvement [57]. Furthermore, in uterine perforation cases, ectopic abdominal pregnancies may result from reimplanting an intrauterine pregnancy while attempting to terminate the pregnancy [58,59].

It was mentioned that there might be an association between uterine perforation and adverse obstetric outcomes. Placenta praevia has been reported to account for 1.4% of patients with a history of uterine rupture, while the rate of placenta praevia in the general population is much lower, at 0.3–0.5% [60]. The need for manual removal of the placenta after vaginal delivery in patients with prior injury of the uterine wall has been documented to be 2.7% [61]. In addition, patients with history of uterine perforation have a higher risk of uterine rupture that must be addressed during delivery [5]. More, some issues related to future fertility should also be communicated to the patient after proper management of uterine perforation with omentum incarceration. All 12 cases in this review reported an uneventful post-operative period and a favorable short-term outcome. Only one paper presented an excellent long-term outcome demonstrated by four subsequent pregnancies that reached full term and resulted in uncomplicated vaginal deliveries [24].

### 5.7. Prevention of Uterine Perforation

All safe intrauterine procedures, including obstetric or non-obstetric D&C, should benefit from a detailed preoperative clinical evaluation and preventive measures [14]. Healthcare providers should assess the risk factors before any gynaecological intervention. They should correctly calculate gestational age to adapt the method of pregnancy termination. Adequate preparation of the uterine cervix is mandatory before any intrauterine manoeuvre with progressive dilation using misoprostol, osmotic or candle dilators [38]. During the intervention, a correct position of the patient and the uterus is necessary as additional preventive measures for the safe use of operative intrauterine instruments.

### 5.8. Strengths and Limitations

Uterine perforation with intra-abdominal evisceration, including omentum involvement, can lead to high maternal morbidity and mortality, especially secondary to termination of pregnancy. Therefore, unsafe abortion is considered a significant public health concern. We believe this thorough review and case report presentation to be a warning sign for this rare but potentially fatal complication. With this paper, we wish to draw attention to a multiplanar approach that should be taken as a matter of urgency after the correct diagnosis of uterine perforation. However, the review has some limitations: the small number of cases because of the rare nature of the condition and the underdiagnoses and underreporting of uterine perforation with omentum incarceration. In addition, all publications, except one, are single case reports that lack certain data.

## 6. Conclusions

All intrauterine procedures should be performed with caution, and ultrasound guidance should be considered, according to the circumstances. Although most uterine perforations are spontaneously resolved, they still represent one of the most severe complications and a source of long-term complications, especially when abdominal viscera is involved. We highlighted the importance of a thorough gynaecological assessment following a D&C procedure that includes a careful clinical examination and a detailed ultrasound evaluation. Healthcare providers should not overlook the diagnosis of omentum involvement in patients with a history of intrauterine procedures, suggestive symptoms or the ultrasound appearance of a hyperechoic endometrial lesion penetrating the uterine wall. The final diagnosis requires a hysteroscopic inspection of the uterine cavity and surgical exploration of the abdominal cavity to pursue the best available management for the best outcome.

## Figures and Tables

**Figure 1 diagnostics-13-00331-f001:**
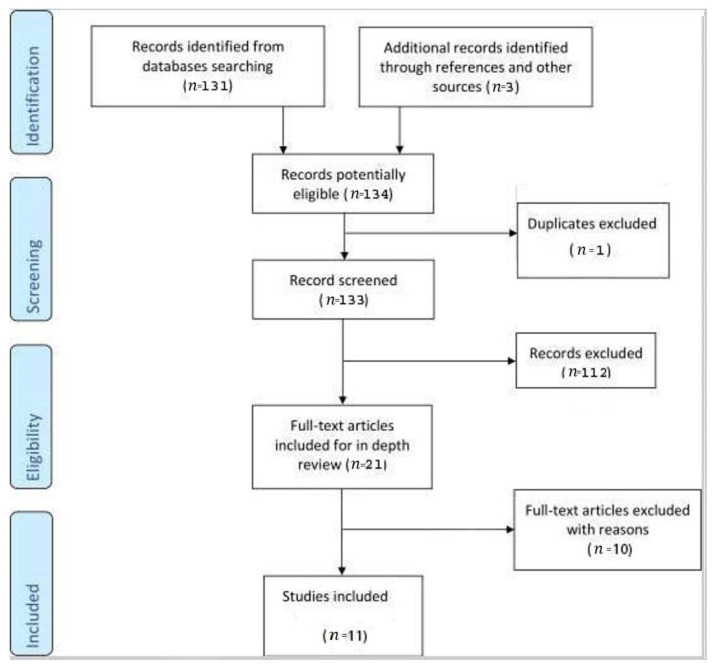
Flowchart illustrating the selection of reports included in the analysis.

**Figure 2 diagnostics-13-00331-f002:**
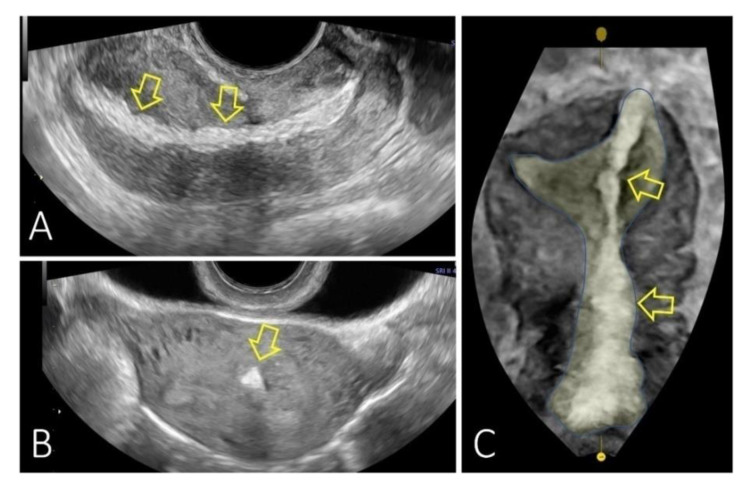
Ultrasound evaluation at 6 h after the curettage. (**A**): Longitudinal view of the uterus with the identification of an echogenic band in the uterine wall and cavity extending from the uterine fundus to the cervical external os (yellow arrows); (**B**): Transversal plane of the uterine body for the width evaluation of the echogenic area; (**C**): 3D reconstruction of the uterine coronal plane showing the endometrial and cervical cavity and the localization of the echogenic area. (Case from the Department of Obstetrics and Gynecology, University of Medicine and Pharmacy of Craiova, Romania).

**Figure 3 diagnostics-13-00331-f003:**
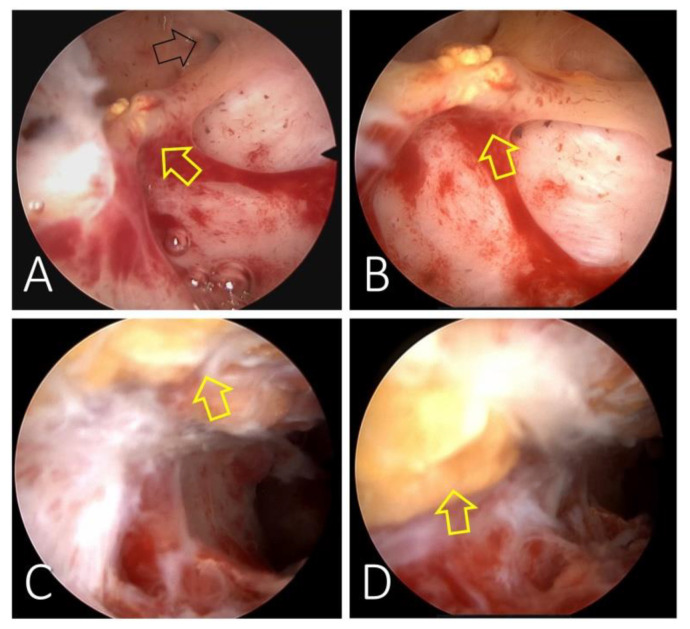
Hysteroscopic evaluation of the uterine cavity at two weeks after the curettage. A band with a fibro-lipomatous appearance (yellow arrows) is identified running from the uterine fundus, next to the tubal ostium (**A**), continuing through the entire endometrial cavity (**B**) to the cervical canal (**C**,**D**).

**Table 1 diagnostics-13-00331-t001:** Characteristics of the included cases.

Authors	Year	Study Type	Age ^1^	Gravida/ Para	Pregnancy Status ^2^	Risk Factors	Imaging	Time from the D&Cto Diagnosis ^3^
Alkhateeb et al. [24]	2015	CR	20	G2P1	13w	Twin pregnancy	US	At the moment of curettage
Chandi et al. [33]	2016	CS	24	G2P1	YES	Unsafe abortion	NO	2 days
			26	G3P2	YES	Abortion	NO	7 h
Myounghwan [25]	2014	CR	26	G2P1	11w	Abortion	US, CT	Immediate after the curettage
Koshiba et al. [26]	2011	CR	31	G4P3	17w	C-S	US, MRI	28 days
La et al. [27]	2021	CR	26	G3P1	YES miscarriage	C-S	US	3 months
Leibner et al. [28]	1995	CR	30	G3P1	first-trimester	Abortion	Rx	17 days
Marsden et al. [29]	1984	CR	25	G4P1	first-trimester	Abortion	NA	Immediate after birth
Nam et al. [30]	2021	CR	57	G2P2	No	D&C	US	28 days
Nayak et al. [31]	2013	CR	32	G3P2	No	Unsafe abortion	US	5 years
Ozaki et al. [34]	2013	LE	28	G2P0	16w	D&C	US	2 years
Sedrati et al. [32]	2022	CR	36	G3P2	NA	D&C for Incomplete Miscarriage	US	7 months

Abbreviations: CR, Case Report; CS, Case Series; LE, Letter to the Editor; US, Ultrasound; C-S, Cesarean Section; MRI, Magnetic Resonance Imaging; NA, Non-Available; Rx, Radiography; D&C, Dilatation and curettage. ^1^ Age in years; ^2^ Pregnancy duration in weeks; ^3^ Known or estimated time from the D&C to the diagnosis.

**Table 2 diagnostics-13-00331-t002:** Clinical data, anamnesis, imaging results (CT-computer tomography, MRI-magnetic resonance imaging, US-ultrasound).

	Symptoms	Anamnesis	History of Intrauterine Applied Procedures	Clinic Examination	Imaging
Alkhateebet al. [24]	- Lower abdominal pain	- 3 months later	- 3 consecutive D&C	- The omental tissue pulled out through the vagina	- Pelvic US: miscarriage 13 weeks of gestation
Chandiet al. [33]	- Vaginal bleeding		- Dai handling following spontaneous incomplete abortion	- Small gut along with omentum coming out of introitus	NA
	- Lower abdominal pain - Vaginal bleeding		- D&C 7hours previous	- The abdomen was soft, and the uterus corresponded 14 weeks in size Omentum was seen coming out through the os	NA
Myounghwan [25]	- Lower abdominal pain		- Uterine perforation during D&C	- Diffuse abdominal tenderness and rebound tenderness	- CT: no evidence of bowel injury except hematoma around the perforation scar
Koshibaet al. [26]	- Lower abdominal pain - Vaginal bleeding		- D&C for a missed abortion		- MRI: fatty mass
La et al. [27]	- Vaginal bleeding - Lower abdominal pain.	- 3 months later	- Two consecutive D&C		- US: omentum embedded into the myometrium suggestive of a previous uterine perforation
Leibneret al. [28]	- Upper abdominal pain - Nausea and vomiting (for two weeks’ duration).	- 1 day later	- Vacuum aspiration termination of pregnancy		- Radiographs of the chest and abdomen—ileus or partial small-bowel obstruction without evidence of free air
Marsden et al. [29]			- 3 consecutive D&C	- Fatty tissue protruding from the cervical os following vaginal delivery	NA
Nam et al. [30]	- Abdominal pain - Menopausal vaginal bleeding	- No regular check-ups - Only Pap smears	- D&C 23 years ago for abnormal uterine bleeding		- US: a hyperechoic round mass with a thick band-like structure penetrating the uterine wall and blood vessels in it on colour Doppler exam
Nayak et al. [31]	- Lower abdominal pain	- Abortion 5 years earlierafter4months of pregnancy		- The foreign body was hanging from the introitus	- US: a tubular and slender foreign body coiled up in the pelvis and probably in the uterine cavity
Ozaki et al. [34]	- Asymptomatic - Referred to a hospital at 16 weeks gestation for a high-risk obstetric consultation	- 2 years later	- D&C		- US: a hyperechogenic structure in the anterior wall of the uterine body with suspected incarceration of the omentum or mesenteric fat
Sedratiet al. [32]	- Amenorrhea - Lower abdominal pain for seven months post-operatively.		- D&C for incomplete miscarriage		- US: discontinuity in the uterine serosa with a hyperechoic mass protruding from the peritoneal cavity into the myometrium suggesting an incarcerated pelvic organ

**Table 3 diagnostics-13-00331-t003:** Applied surgical approach, intraoperative findings and management.

	Surgical Approach	Intraoperative Findings	Management
Alkhateeb et al. [24]	- Laparotomy	- Uterine perforation at the fundus with the omentum pulled in through the perforation	- The omentum was drawn out of the uterus, transfixed, ligated by suture and trimmed. - Uterus perforation was sutured.
Chandi et al. [33]	Case 1	- A rent of 7 × 3 cm was present in the lower uterine segment’s anterior wall of the uterus.	- Resection of the 20 cm of ileum and caecum was done, and ileo-ascending colon end-to-end anastomosis was performed. - 2 units of whole blood and 1 unit of FFP were transfused intraoperatively, and two units of FFP post-operatively. - Uterus perforation was sutured.
Chandi et al. [33]	Case 2	- Hemoperitoneum of 200 cm^3^ - A rent of 5 cm was present in the anterior uterine wall in the lower uterine segment extending to the left laterally and downwards to the vagina. - Utero-vesical pouch was already breached. - The bladder wall was intact. -Fetal skull was removed from the UV pouch. - B/L tubes and ovaries were standard. - The gut and bladder were normal.	
Myounghwan [25]	- Laparoscopy	- Perforation scar of the uterine fundus	- Incarcerated omentum was incarcerated. - Suture at the perforation site - 4 units of packed red blood cells were transfused.
Koshiba et al. [26]	- Laparotomy	- Uterine perforation distant from the previous cesarean scar	- Dissection of the omental loop. - Uterine perforation was sutured.
La et al. [27]	- Laparoscopy	- Fundal defect	- Omentum was released. - The uterus defect was sutured.
Leibner et al. [28]	- Laparotomy	- 2 perforations of the body of the uterus 1 cm (one contained herniated omentum). - A strangulated 5-cm segment of the extrauterine small bowel with complete obstruction at this level.	- The ischemic segment of the bowel was resected with immediate end-to-end anastomosis. - The uterus was not repaired.
Marsden et al. [29]	- Laparotomy	- A portion of the greater omentum passed into the myometrium at the right corm of the uterus.	- Gentle traction was used to remove the omentum from the uterine cavity. - Uterine perforation was sutured.
Nam et al. [30]	- Office hysteroscopy	- A pale-yellowish mass with intrauterine adhesions was observed.	- Laparoscopic and hysteroscopic resection of the incarcerated omentum. - Uterine perforation was sutured.
	- Laparoscopy	- An incarcerated omentum into the fundus of the uterine cavity through the uterine perforation site was noticed	
Nayak et al. [31]	- Cystoscopy	- Excluded bladder involvement	
	- Hysteroscopy	- Showed that the tube had pierced through the posterior wall of the uterus - There were no intrauterine adhesions.	
	- Laparoscopy	- Ryle’s tube had perforated the uterus through the posterior fundal wall. - Bowel and omental loops were adherents to the entire length of the intra-abdominal portion of the tube.	
	- Laparotomy		- Adhesiolysis and the freed tube was dragged out vaginally - Suture of the uterine fundus perforation -Bilateral tubectomy
Ozaki et al. [34]	-C-S	- Omentum was incarcerated in the anterior wall of the uterine body	- The omental loop was dissected.
Sedrati et al. [32]	- Hysteroscopy	- Severe intrauterine adhesions	- The omentum was excised.
	- Laparoscopy	- Omental incarceration	- The uterine serosa was sutured.

## Data Availability

All the studies used in this study are published in the literature.

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
