# Peer review of "Uterine Perforation as a Complication of the Intrauterine Procedures Causing Omentum Incarceration: A Review"

_diagnostics, 2023, doi:10.3390/diagnostics13020331_

Round 1

Reviewer 1 Report

I suggest to delete Figure 4 becouse of a low scientefic priority of it.

Author Response

#1. I suggest to delete Figure 4 because of a low scientefic priority of it.

Reply #1. We deleted the figure, as it is not informative regarding the subject of the review.

Reviewer 2 Report

The authors present a case report and review of literature about a very rare event - incarceration of omentum after uterine perforation. As said by the authors, uterine perforation is by itself a rare event. The manuscript is interesting especially for general gynecologists.

The methods are clearly described. I would however like the authors to explain their exclusion criteria (presence of the malignancy, D&C after delivery and perforation after insertion of IUD). I do not understand why these conditions should not be incorporated in the review. This should be explained or included in the review. 

In general the manuscript has some redundancy. It could be shortened and more concise. 

The conclusions could be modified. The authors state that the uterine perforation is severe. On the other hand they presented that most uterine perforations are spontaneously resolved and do not need surgical intervention. 

The language is generally understandable for a non native speaker although there are some sections that are not very clear. English editing is therfore recommended.

Author Response

#1. The authors present a case report and review of literature about a very rare event - incarceration of omentum after uterine perforation. As said by the authors, uterine perforation is by itself a rare event. The manuscript is interesting especially for general gynecologists.

Reply#1. We thank the reviewer for his/her appreciation.

#2. The methods are clearly described. I would however like the authors to explain their exclusion criteria (presence of the malignancy, D&C after delivery and perforation after insertion of IUD). I do not understand why these conditions should not be incorporated in the review. This should be explained or included in the review. 

Reply #2. The reviewer observation is correct. We decided not to include the conditions that represent risk factors for uterine perforation, because in such cases the professionals are well-aware of the potential complications. Instead, we aimed to describe the diagnosis and outcome of uterine perforation with omentum incarceration in low-risk women, where the expectances for such complications is low and the diagnosis can be easily overlooked.

We included this paragraph in the Methods section to better explain the selection criteria for the review, pages 2-3, lines 96-101.

#3. In general the manuscript has some redundancy. It could be shortened and more concise. 

Reply #3. We thank the reviewer for pointing out this aspect. The manuscript went through extensive changes that addressed also this matter. All the changes can be easily identified, as track changes function was activated.

#4. The conclusions could be modified. The authors state that the uterine perforation is severe. On the other hand they presented that most uterine perforations are spontaneously resolved and do not need surgical intervention. 

Reply #4. The observation is correct, the statement may be misleading. We decided to modify the paragraph, that now reads: “Although most uterine perforations are spontaneously resolved, it still represents one of the most severe complication and a source for long-term complications, especially when abdominal viscera is involved.” (page 13, lines 398-400).

#5. The language is generally understandable for a non native speaker although there are some sections that are not very clear. English editing is therefore recommended.

Reply #5. We thank the reviewer for pointing out this aspect. The manuscript went through extensive changes that mainly addressed language editing. Track changes function was activated to highlight all the changes that have been made.

Round 2

Reviewer 2 Report

I would like to thank the authors for considering the comments. I believe the manuscript is much more clear.